# The Association between Urinary Diversion Type and Other-Cause Mortality in Radical Cystectomy Patients

**DOI:** 10.3390/cancers16020429

**Published:** 2024-01-19

**Authors:** Simone Morra, Lukas Scheipner, Andrea Baudo, Letizia Maria Ippolita Jannello, Mario de Angelis, Carolin Siech, Jordan A. Goyal, Nawar Touma, Zhe Tian, Fred Saad, Gianluigi Califano, Massimiliano Creta, Giuseppe Celentano, Shahrokh F. Shariat, Sascha Ahyai, Luca Carmignani, Ottavio de Cobelli, Gennaro Musi, Alberto Briganti, Felix K. H. Chun, Nicola Longo, Pierre I. Karakiewicz

**Affiliations:** 1Cancer Prognostics and Health Outcomes Unit, Division of Urology, University of Montréal Health Center, Montréal, QC H2X 3E4, Canada; l.scheipner@medunigraz.at (L.S.); andrea.baudo@unimi.it (A.B.); letizia.jannello@unimi.it (L.M.I.J.); deangelis.mario@hsr.it (M.d.A.); caroli.siech@kgu.de (C.S.); jordan.goyal@umontreal.ca (J.A.G.); nawar.touma@umontreal.ca (N.T.); tian.zhe@umontreal.ca (Z.T.); fred.saad@umontreal.ca (F.S.); pierre.karakiewicz@umontreal.ca (P.I.K.); 2Department of Neurosciences, Science of Reproduction and Odontostomatology, University of Naples Federico II, 80131 Naples, Italy; gianluigi.califano@unina.it (G.C.); massimiliano.creta@unina.it (M.C.); giuseppe.celentano@unina.it (G.C.); nicola.longo@unina.it (N.L.); 3Department of Urology, Medical University of Graz, 8010 Graz, Austria; sascha.ahyai@medunigraz.at; 4Department of Urology, IRCCS Policlinico San Donato, 20097 Milan, Italy; luca.carmignani@unimi.it; 5Department of Urology, IEO European Institute of Oncology, IRCCS, Via Ripamonti 435, 20141 Milan, Italy; decobelli.ottavio@ieo.it (O.d.C.); gennaro.musi@ieo.it (G.M.); 6Department of Urology, Università degli Studi di Milano, 20126 Milan, Italy; 7Division of Experimental Oncology, Unit of Urology, Urological Research Institute (URI), IRCCS San Raffaele Scientific Institute, 20132 Milan, Italy; briganti.alberto@hsr.it; 8Department of Urology, University Hospital Frankfurt, Goethe University Frankfurt am Main, 39120 Frankfurt am Main, Germany; felix.chun@kgu.de; 9Department of Urology, Comprehensive Cancer Center, Medical University of Vienna, 1090 Vienna, Austria; shahrokh.shariat@meduniwien.ac.at; 10Department of Urology, Weill Cornell Medical College, New York, NY 10065, USA; 11Department of Urology, University of Texas Southwestern Medical Center, Dallas, TX 75390, USA; 12Hourani Center of Applied Scientific Research, Al-Ahliyya Amman University, Amman 19328, Jordan; 13Department of Urology, IRCCS Ospedale Galeazzi—Sant’Ambrogio, 20157 Milan, Italy; 14Department of Oncology and Haemato-Oncology, Università Degli Studi di Milano, 20122 Milan, Italy

**Keywords:** radical cystectomy, urinary diversion, orthotopic neobladder, ileal conduit, abdominal pouch, other-cause mortality

## Abstract

**Simple Summary:**

This study, conducted within a large North American cohort from the Surveillance, Epidemiology and End Results (SEER) database (2004–2020), aimed to investigate whether more complex urinary diversion (UD) procedures, such as orthotopic neobladder and abdominal pouch, are associated with higher other-cause mortality (OCM) compared to the conventional ileal conduit in T1-T4aN0M0 bladder cancer patients. Among 3008 patients, 79% underwent ileal conduit, while 21% opted for continent UD. After rigorous analysis, including propensity score matching and multivariable adjustments, the study found that neither continent UD nor its subtypes (orthotopic neobladder and abdominal pouch) were associated with higher 10-year OCM rates relative to ileal conduit. The conclusion suggests that more intricate UD procedures do not seem to elevate OCM risk compared to the simpler ileal conduit.

**Abstract:**

Background: It is unknown whether more complex UD, such as orthotopic neobladder and abdominal pouch, may be associated with higher OCM rates than ileal conduit. We addressed this knowledge gap within the SEER database 2004–2020. Methods: All T1-T4aN_0_M_0_ radical cystectomy (RC) patients were identified. After 1:1 propensity score matching (PSM), cumulative incidence plots, univariable and multivariable competing-risks regression (CRR) models were used to test differences in OCM rates according to UD type (orthotopic neobladder vs. abdominal pouch vs. ileal conduit). Results: Of all 3008 RC patients, 2380 (79%) underwent ileal conduit vs. 628 (21%) who underwent continent UD (268 orthotopic neobladder and 360 abdominal pouch). After PSM relative to ileal conduit, neither continent UD (13 vs. 15%; *p* = 0.1) nor orthotopic neobladder (13 vs. 16%; *p* = 0.4) nor abdominal pouch (13 vs. 15%; *p* = 0.2) were associated with higher 10-year OCM rates. After PSM and after adjustment for cancer-specific mortality (CSM), as well as after multivariable adjustments relative to ileal conduit, neither continent UD (Hazard Ratio [HR]:0.73; *p* = 0.1), nor orthotopic neobladder (HR:0.84; *p* = 0.5) nor abdominal pouch (HR:0.77; *p* = 0.2) were associated with higher OCM. Conclusions: It appears that more complex UD types, such as orthotopic neobladder and abdominal pouch are not associated with higher OCM relative to ileal conduit.

## 1. Introduction

Bladder cancer is the 10th most commonly diagnosed cancer worldwide in both sexes [1]. According to the latest European Association of Urology (EAU) guidelines, radical cystectomy (RC) with urinary diversion (UD), continent or ileal conduit (IC), is the standard of care for T2-T4a N0M0 bladder cancer patients [2]. In select patients, continent UD, especially orthotopic neobladder, provides the closest bladder function to that of a native bladder [3]. Although continent UD has body image and body function benefits over IC [4,5,6], its greater complexity may also predispose to medical and/or surgical complications [7], beyond those of IC, which in term may predispose to higher other-cause mortality (OCM).

The above hypothesis has never been tested. We addressed this knowledge gap and tested for OCM rate differences between continent UD and IC, where OCM represents the end marker of comorbidities that may undermine survival. Our tests first addressed the comparison between all continent UD and IC patients, in addition to more focused analyses that specifically addressed orthotopic neobladder and then abdominal pouch patients. To address this knowledge gap, we relied on a contemporary large-scale population-based database, Surveillance, Epidemiology and End Results (SEER).

## 2. Materials and Methods

### 2.1. Study Population

The SEER database samples 34.6% of the United States population in terms of demographic composition and cancer incidence [8]. Within the SEER database from 2004 to 2020, we identified patients ≥ 18 years old with newly diagnosed and histologically confirmed T1-T4aN0M0 urothelial bladder carcinoma (International Classification of Disease for Oncology [ICD-O-3] site code C67.0–67.9) [9]. We included only patients ≤ 70 years old who underwent radical RC with known UD type (orthotopic neobladder, abdominal pouch or ileal conduit). All autopsy or death certificate-only cases, patients with missing follow-up, missing vital status and unknown data regarding grade or marital status were excluded. Since the SEER database is entirely anonymous, study-specific ethics approval was waived by the institutional review board.

### 2.2. Variables and Outcome of Interest

For each patient, we recorded UD type (continent vs. IC). Subsequent stratification among continent UD patients was made between orthotopic neobladder and abdominal pouch. Other variables included age at diagnosis (years), sex, T stage (T1 vs. T2 vs. T3 vs. T4a), grade (low vs. high), systemic therapy status (yes vs. no), race/ethnicity (Caucasian vs. other) and marital status (married vs. unmarried). The primary study endpoint consisted of OCM, which was defined using the SEER code for cause of death.

### 2.3. Statistical Analyses

Four analytical steps were completed. First, we tabulated baseline patient and tumor characteristics. Second, 1:1 propensity score matching (PSM) according to the nearest neighbor was applied between continent UD and IC. Variables in PSM included age at diagnosis, sex, race/ethnicity, marital status, T stage, grade and systemic therapy status to maximally reduce the effect of bias and confounding [10]. Third, cumulative incidence plots depicted OCM rates, after accounting for cancer-specific mortality (CSM). Subsequently, univariable and multivariable competing risks regression (CRR) models tested whether UD type (continent vs. IC) independently predicts OCM after adjustment for CSM. Fourth, the same methodological steps (1:1 PSM, cumulative incidence plots and CRR models) were repeated in two subgroup analyses: (1) orthotopic neobladder vs. IC and (2) abdominal pouch vs. IC. All tests were two-sided, with a significance level set at *p* < 0.05. In all statistical analyses, R software environment for statistical computing and graphics (R version 4.1.3, R Foundation for Statical Computing, Vienna Austria; http://www.r-project.org/, accessed on 14 January 2023) was used [11].

## 3. Results

### 3.1. Descriptive Characteristics of RC Patients Treated with Continent UD vs. IC

Of all 3008 RC patients, 628 (21%) underwent continent UD vs. 2380 (79%) who underwent IC (Table 1). Relative to IC, continent UD patients were younger at diagnosis (60 vs. 63 years; *p* < 0.001), more often male (91 vs. 84%; *p* < 0.001) and married (72 vs. 65%: *p* < 0.001), more frequently harbored localized (T1-T2) disease (72 vs. 65%; *p* = 0.003) and were less frequently exposed to systemic therapy (40 vs. 45%; *p* = 0.03). No statistically significant or clinically meaningful differences were recorded for race/ethnicity (*p* = 0.1) or grade distribution (*p* = 0.6).

After 1:1 PSM (continent UD vs. IC), 619 (99%) of 628 continent UD patients were matched with 619 (25%) of 2380 IC patients. No statistically significant residual differences remained in age, sex, marital status, T stage and systemic therapy status between the two groups.

### 3.2. Survival Outcomes of RC Patients Treated with Continent UD vs. IC

After 1:1 PSM, 10-year OCM rates were 13% in continent UD vs. 15% in IC patients (*p* = 0.1). In univariable CRR models, continent UD was not associated with higher OCM than IC (Hazard Ratio [HR]:0.79, 95% Confidence Interval [CI]: 0.59–1.06; *p* = 0.1). Moreover, in multivariable CRR models, continent UD was not an independent predictor of higher OCM (HR:0.80, 95% CI: 0.60–1.07; *p* = 0.1) after adjustment for CSM and after multivariable adjustments for age, sex, race/ethnicity, marital status, T stage, systemic therapy status and grade (Table 2).

### 3.3. Descriptive Characteristics and Survival Outcomes of RC Patients Treated with Orthotopic Neobladder vs. Ileal Conduit

Of all 2648 RC patients, 268 (10%) underwent orthotopic neobladder vs. 2380 (90%) who underwent ileal conduit (Table 3). After 1:1 PSM (orthotopic neobladder vs. ileal conduit), 263 (98%) of 268 orthotopic neobladder patients were matched with 263 (11%) of 2380 ileal conduit patients. No statistically significant residual differences remained in age, sex, race/ethnicity, marital status, T stage, grade or systemic therapy status between the two groups.

After 1:1 PSM, 10-year OCM rates were 13% in orthotopic neobladder patients vs. 16% in ileal conduit patients (*p* = 0.4). In univariable CRR models, orthotopic neobladder was not associated with higher OCM than ileal conduit (HR:0.82, 95% CI: 0.53–1.28; *p* = 0.4). Moreover, in multivariable CRR models, orthotopic neobladder was not an independent predictor of higher OCM (HR:0.84, 95% CI: 0.54–1.31; *p* = 0.5) after adjustment for CSM and after multivariable adjustments for age, sex, race/ethnicity, marital status, T stage and systemic therapy status (Table 4).

### 3.4. Descriptive Characteristics and Survival Outcomes of RC Patients Treated with Abdominal Pouch vs. Ileal Conduit

Of all 2740 RC patients, 360 (13%) underwent abdominal pouch vs. 2380 (87%) who underwent ileal conduit (Table 5). After 1:1 PSM (abdominal pouch vs. ileal pouch), 355 (99%) of 360 abdominal pouch patients were matched with 355 (15%) of 2380 ileal conduit patients. No statistically significant residual differences remained in age, sex, race/ethnicity, marital status, T stage, grade or systemic therapy status between the two groups.

After 1:1 PSM, 10-year OCM rates were 13% in abdominal pouch patients vs. 15% in ileal conduit patients (*p* = 0.2). In univariable CRR models, abdominal pouch was not associated with higher OCM than ileal conduit (HR:0.78, 95% CI: 0.52–1.15; *p* = 0.2). Moreover, in multivariable CRR models, abdominal pouch was not an independent predictor of higher OCM (HR:0.77, 95% CI: 0.52–1.14; *p* = 0.2), after adjustment for CSM and after multivariable adjustments for age, sex, race/ethnicity, marital status, T stage and systemic therapy status (Table 6).

## 4. Discussion

Due to the greater complexity of continent UD, we hypothesized that it may predispose to higher rates of medical and/or surgical complications [7], beyond those of IC. Such complications may in turn undermine life expectancy and result in higher OCM rates in continent UD, which included orthotopic neobladder and abdominal pouch relative to IC. We tested the above hypothesis within a large-scale population-based cohort and made several noteworthy observations.

First, continent UD is rare, relative to IC. Specifically, in the current study of all RC patients, only 628 (21%) underwent continent UD vs. 2380 (79%) who underwent IC. Of all 628 continent UD patients, 268 (43%) underwent orthotopic neobladder vs. 360 (57%) who underwent abdominal pouch. These rates are comparable to previous reports. For example, Nahar et al. (National Cancer Database [NCDB] 2004–2013) recorded 11,933 RC patients over a period of 10 years, of whom 1736 (15%) underwent continent UD vs. 10,197 (85%) who underwent IC. Of all 1736 continent UD patients, 692 (40%) underwent orthotopic neobladder vs. 1044 (60%) who underwent abdominal pouch [12]. These results indicate that large-scale databases, such as SEER or NCDB, are required to assess survival outcomes in continent UD patients, especially when mature follow-up is needed. Indeed, it is noteworthy that in the current study, the median follow-up was as long as 10.1 years, including only contemporary diagnosed patients (2004–2020). To the best of our knowledge, only Eisenberg et al. (Mayo Clinic 1980–2006) described equally mature follow-up (10.5 years) in RC patients [13]. However, Eisenberg et al. predominantly relied on RC patients diagnosed prior to the year 2000. Such practice may invalidate the generalizability of observations to contemporary RC patients. In consequence, the current study offers mature follow-up (10.1 years) in addition to relying on exclusively contemporary RC patients (2004–2020) who are reflective of surgical UD types that are in use today. To the best of our knowledge, other large-scale RC series relied on more historical cohorts and on shorter follow-up. For example, Takahashi et al. retrospectively identified in a multi-institutional pooled analysis from 21 centers over a time span of five years (1991–1995) 518 RC patients with a median follow-up of 4.4 years [14]. Similarly, Chromecki et al. retrospectively identified in a multi-institutional database from 12 centers over a time span of 30 years (1979–2008) 4118 RC patients with a median follow-up of 3.7 years [15].

Second, several previous studies reported on large-scale RC patient cohorts [16,17,18,19]. Moreover, such studies also tested for differences according to UD type [18,19]. However, no study examined OCM rates according to UD type: continent vs. IC or orthotopic neobladder vs. IC or abdominal pouch vs. IC. We addressed this knowledge gap based on the notion that all types of continent UD may predispose to higher rates of medical and/or surgical complications. Such complications may in turn undermine life expectancy and result in higher rates of OCM, which represented the primary study endpoint. Since important differences are known to distinguish continent UD vs. IC patients, which could bias or confound OCM rates, several statistical methods attempted to maximally reduce the effect of uncontrol bias or confounders. Specifically, we relied on 1:1 PSM. Matching variables included patient characteristics (age at diagnosis, sex, race/ethnicity, marital status) and tumor characteristics (T stage, grade), as well as systemic therapy status. Additional multivariable adjustment was made to account for potential residual differences in all the above variables. Finally, to reflect the selection criteria for continent UD, analyses exclusively focused on patients ≤ 70 years old at diagnosis, since continent UD is rarely offered to older patients. Moreover, older patients may succumb to higher rates of OCM due to the effect of advanced age and comorbidities that are otherwise unrelated to UD type.

Third, OCM rate comparisons of continent vs. IC, as well as of orthotopic neobladder vs. IC and finally of abdominal pouch vs. IC, all revealed an absence of statistically significant differences in univariable as well as in multivariable CRR models that were adjusted for CSM as well as for the residual effect of differences in all covariates after 1:1 PSM for all such covariates. The lack of OCM differences in the overall comparison between continent vs. IC, as well as in the two additional comparisons of orthotopic neobladder vs. IC and abdominal pouch vs. IC, convincingly demonstrated that the increased surgical complexity of either continent UD vs. IC is not a risk factor for medical and/or surgical complications that may undermine life expectancy. This observation is noteworthy since it may reassure patients about the safety and innocuity of continent UD at pre-operative counseling and decision-making about UD type. Our observations may potentially motivate some patients who are otherwise cautious about choosing a more surgically challenging and complex reconstruction, relative to IC. Our results cannot be directly compared to previous studies addressing OCM after RC according to different UD type, since no such studies have been performed. It is also noteworthy that OCM rates after RC have not been reported regardless of UD type, with or without a comparison to a reference group. In consequence, we provide novel observations that will be of use to clinicians and patients.

Despite its strengths, the current study is not devoid of limitations. First, despite a large patient population, the amount of detail is limited. Indeed, only OCM rates were available. Although OCM represents the ultimate marker of life-threatening comorbidities and complications, other endpoints would have been of interest. For example, short-term, mid-term and long-term medical and/or surgical complications according to UD type would ideally be available for the purpose of further adjustment. Second, baseline comorbidity status, such as smoking status, body mass index (BMI), diabetes mellitus status, hypertension, cardiovascular disease and the American Society of Anaesthesiologists (ASA) score, as well as ECOG status or the Charlson Comorbidity Index (CCI) of RC patients, was also not available. Ideally, it could have been used for the purpose of further adjustment. The lack of inclusion of medical and/or surgical complications, as well as the lack of consideration of baseline comorbidity status, preclude the generalizability of our OCM observation regarding earlier adverse outcomes, such as renal insufficiency, infections, gastrointestinal complications and others that can affect RC patients. Third, adjustments could not be made for patient and surgeon preferences that underlie decision-making regarding UD type. This limitation is applicable to all studies that compare continent UD and IC [7,18,19]. A randomized design that is free of various biases, such as those described above, as well as of the confounding of patient and/or surgeon preferences regarding UD type at RC, cannot be expected to ever be completed [7,18,19]. Fourth, the level of detail for systemic therapy is limited. Indeed, the SEER database does not have the granularity to allow the identification of specific systemic therapy regimens (chemotherapy or immunotherapy) and does not provide information on cycle number and systemic therapy duration. Fifth, SEER does not provide any information regarding the surgical approach (open vs. laparoscopic vs. robotic). Indeed, several investigators reported differences in terms of quality of life (QoL) according to surgical approach [20], as well as in the rate of early and late complications that may affect OCM rates [21,22,23]. Sixth, we included only patients ≤ 70 years old. The selection was made in order to minimize differences in comorbidities in the different groups that may be associated with different OCM rates. Additionally, according to EAR guidelines, orthotopic neobladder should be discouraged in patients older than 80 years old [2]. Moreover, several previous investigators who focused on differences in clinical outcomes according to UD that included orthotopic neobladder recorded a median age between 60 and 70 years old [4,7,24,25,26,27,28,29,30]. Therefore, the selection made in the current study reflects the treatment options applied in practice in daily life. Finally, the current study shares the limitations of all similar studies that were based on the SEER database and relied on a retrospective data design [12,18].

## 5. Conclusions

According to different UD types (continent vs. IC), no differences in OCM rates were recorded. Moreover, even when comparisons were made of orthotopic neobladder vs. IC and abdominal pouch vs. IC, no differences in OCM rates were recorded.

## Figures and Tables

**Table 1 cancers-16-00429-t001:** Descriptive characteristics of newly diagnosed urothelial bladder cancer patients (T1-T4aN0M0) treated with radical cystectomy and urinary diversion (UD) within Surveillance, Epidemiology, and End Results (SEER) 2004–2020, before and after 1:1 propensity score matching (PSM). Results are stratified according to different UD types (continent UD vs. ileal conduit).

	Before PSM				After PSM 1:1			
Characteristic	Overall, *n* = 3008 ^1^	Continent UD, *n* = 628 (21%) ^1^	Ileal Conduit, *n* = 2380 (79%) ^1^	*p*-Value ^2^	Overall, *n* = 1238 ^1^	Continent UD, *n* = 619 (50%) ^1^	Ileal Conduit, *n* = 619 (50%) ^1^	*p*-Value ^2^
**Age**	63 (57, 67)	60 (55, 66)	63 (58, 67)	**<0.001**	61 (55, 66)	61 (55, 66)	61 (56, 66)	0.4
**Male sex**	2563 (85%)	573 (91%)	1990 (84%)	**<0.001**	1136 (92%)	564 (91%)	572 (92%)	0.4
**Caucasian**	2456 (82%)	526 (84%)	1930 (81%)	0.1	1043 (84%)	518 (84%)	525 (85%)	0.6
**Married**	2000 (66%)	455 (72%)	1545 (65%)	**<0.001**	893 (72%)	447 (72%)	446 (72%)	0.9
**T stage**				**0.003**				0.7
T1	481 (16%)	121 (19%)	360 (15%)		244 (20%)	115 (19%)	129 (21%)	
T2	1511 (50%)	330 (53%)	1181 (50%)		648 (52%)	327 (53%)	321 (52%)	
T3	727 (24%)	129 (21%)	598 (25%)		256 (21%)	129 (21%)	127 (21%)	
T4a	289 (10%)	48 (7%)	241 (10%)		90 (7.3%)	48 (7.8%)	42 (6.8%)	
**High grade**	2906 (97%)	609 (97%)	2297 (97%)	0.6	1206 (97%)	600 (97%)	606 (98%)	0.3
**Systemic therapy performed**	1326 (44%)	252 (40%)	1074 (45%)	**0.03**	491 (40%)	250 (40%)	241 (39%)	0.6

^1^ Median (IQR); *n* (%); ^2^ Wilcoxon rank sum test; Pearson’s Chi-square test; UD: urinary diversion.

**Table 2 cancers-16-00429-t002:** Competing risk regression analyses addressing other-cause mortality (OCM) of newly diagnosed urothelial bladder cancer patients (T1-T4aN0M0) treated with radical cystectomy plus urinary diversion within Surveillance, Epidemiology and End Results (SEER) 2004–2020, after adjustment for cancer-specific mortality (CSM).

	OCM					
	Univariable			Multivariable		
	HR ^1^	95% CI	*p*-Value	HR ^1^	95% CI	*p*-Value
**Ileal Conduit**	-	-	*Ref*	-	-	*Ref*
Continent UD	0.79	0.59–1.06	0.1	0.80	0.6–1.07	0.1
**Age**	1.05	1.03–1.08	**<0.001**	1.05	1.03–1.08	**<0.001**
**Male**	-	-	*Ref*	-	-	*Ref*
Female	0.79	0.39–1.60	0.5	0.80	0.38–1.69	0.6
**Caucasian**	-	-	*Ref*	-	-	*Ref*
Non-Caucasian	0.85	0.54–1.31	0.5	0.86	0.55–1.35	0.52
**Married**	-	-	*Ref*	-	-	*Ref*
Unmarried	1.04	0.75–1.44	0.8	1.10	0.79–1.53	0.6
**T1**	-	-	*Ref*	-	-	*Ref*
T2	1.06	0.72–1.57	0.8	1.12	0.76–1.67	0.6
T3	0.93	0.58–1.50	0.8	1.05	0.65–1.70	0.9
T4a	1.29	0.71–2.36	0.4	1.37	0.74–2.56	0.3
**Low grade**	-	-	*Ref*	-	-	*Ref*
High grade	1.00	0.45–2.22	0.9	0.86	0.39–1.89	0.7
**Systemic therapy performed**	-	-	*Ref*	-	-	*Ref*
No	1.08	0.79–1.48	0.6	1.11	0.80–1.53	0.5

^1^ HR = Hazard Ratio, CI = Confidence Interval.

**Table 3 cancers-16-00429-t003:** Descriptive characteristics of newly diagnosed (2004–2020) T1-T4aN0M0 bladder cancer patients treated with radical cystectomy and urinary diversion (UD), before and after the propensity score matching (1:1), within the Surveillance, Epidemiology, and End Results (SEER), according to different UD type (orthotopic neobladder vs. ileal conduit).

	Before PSM				After PSM 1:1			
Characteristic	Overall, *n* = 2648 ^1^	Orthotopic Neobladder, *n* = 268 (10%) ^1^	Ileal Conduit, *n* = 2380 (90%) ^1^	*p*-Value ^2^	Overall, *n* = 526 ^1^	Orthotopic Neobladder, *n* = 263 (50%) ^1^	Ileal Conduit, *n* = 263 (50%) ^1^	*p*-Value ^2^
**Age**	63 (58, 67)	61 (55, 66)	63 (58, 67)	**<0.001**	62 (56, 66)	62 (55, 66)	62 (56, 66)	0.7
**Male sex**	2248 (85%)	258 (96%)	1990 (84%)	**<0.001**	506 (96%)	253 (96%)	253 (96%)	0.9
**Caucasian**	2157 (81%)	227 (85%)	1930 (81%)	0.2	447 (85%)	222 (84%)	225 (86%)	0.8
**Married**	1742 (66%)	197 (74%)	1545 (65%)	**0.005**	382 (73%)	192 (73%)	190 (72%)	0.8
**T stage**				**0.048**				0.9
T1	405 (15%)	45 (17%)	360 (15%)		86 (16%)	43 (16%)	43 (16%)	
T2	1332 (50%)	151 (56%)	1181 (50%)		303 (58%)	148 (56%)	155 (59%)	
T3	652 (25%)	54 (20%)	598 (25%)		101 (19%)	54 (21%)	47 (18%)	
T4a	259 (10%)	18 (7%)	241 (10%)		36 (7%)	18 (7%)	18 (7%)	
**High grade**	2559 (97%)	262 (98%)	2297 (97%)	0.3	517 (98%)	257 (98%)	260 (99%)	0.5
**Systemic therapy performed**	1182 (45%)	108 (40%)	1074 (45%)	0.1	212 (40%)	106 (40%)	106 (40%)	0.6

^1^ Median (IQR); *n* (%); ^2^ Wilcoxon rank sum test; Pearson’s Chi-square test.

**Table 4 cancers-16-00429-t004:** Competing risk regression analyses addressing other-cause mortality (OCM) of newly diagnosed urothelial bladder cancer patients (T1-T4aN0M0) treated with radical cystectomy plus urinary diversion (UD) within Surveillance, Epidemiology and End Results (SEER) 2004–2020, after adjustment for cancer-specific mortality (CSM).

	OCM					
	Univariable			Multivariable		
	HR ^1^	95% CI	*p*-Value	HR ^1^	95% CI	*p*-Value
**Ileal conduit**	-	-	*Ref*	-	-	*Ref*
Orthotopic neobladder	0.82	0.53–1.28	0.4	0.84	0.54–1.31	0.5
**Age**	1.07	1.03–1.1	**<0.001**	1.07	1.03–1.1	**<0.001**
**Male**			*Ref*			*Ref*
Female	1.8	0.60–5.46	0.3	1.74	0.59–5.17	0.3
**Caucasian**			*Ref*			*Ref*
Other	0.67	0.32–1.4	0.3	0.70	0.33–1.45	0.3
**Married**			*Ref*			*Ref*
Unmarried	1.13	0.70–1.83	0.6	1.09	0.64–1.83	0.8
**T1**			*Ref*			
T2	1.93	0.92–4.05	0.08	1.92	0.86–4.28	0.1
T3	1.44	0.61–3.42	0.4	1.54	0.63–3.76	0.4
T4a	2.03	0.7–5.89	0.2	2.02	0.66–6.23	0.2
**Systemic therapy performed**			*Ref*			*Ref*
No	1.01	0.63–1.61	0.9	1.00	0.60–1.65	0.9

^1^ HR: Hazard Ratio; CI: Confidence Interval.

**Table 5 cancers-16-00429-t005:** Descriptive characteristics of newly diagnosed (2004–2020) T1-T4aN0M0 bladder cancer patients treated with radical cystectomy and urinary diversion (UD), before and after the propensity score matching (1:1) within the Surveillance, Epidemiology and End Results (SEER), according to different UD type (abdominal pouch vs. ileal conduit).

	Before PSM				After PSM 1:1			
Characteristic	Overall, *n* = 2740 ^1^	Abdominal Pouch, *n* = 360 (13%) ^1^	Ileal Conduit, *n* = 2380 (87%) ^1^	*p*-Value ^2^	Overall, *n* = 710 ^1^	Abdominal Pouch, *n* = 355 (50%) ^1^	Ileal Conduit, *n* = 355 (50%) ^1^	*p*-Value ^2^
**Age**	63 (58, 67)	60 (55, 65)	63 (58, 67)	**<0.001**	60 (55, 65)	60 (55, 65)	60 (55, 66)	0.7
**Male sex**	2305 (84%)	315 (88%)	1990 (84%)	0.06	630 (89%)	310 (87%)	320 (90%)	0.2
**Caucasian**	2229 (81%)	299 (83%)	1930 (81%)	0.4	585 (82%)	295 (83%)	290 (82%)	0.6
**Married**	1803 (66%)	258 (72%)	1545 (65%)	**0.01**	521 (73%)	254 (72%)	267 (75%)	0.3
**T stage**				**0.02**				0.8
T1	436 (16%)	76 (21%)	360 (15%)		142 (20%)	73 (21%)	69 (19%)	
T2	1360 (50%)	179 (50%)	1181 (50%)		362 (51%)	178 (50%)	184 (52%)	
T3	673 (25%)	75 (21%)	598 (25%)		152 (21%)	74 (21%)	78 (22%)	
T4a	271 (10%)	30 (8%)	241 (10%)		54 (8%)	30 (9%)	24 (7%)	
**High grade**	2644 (96%)	347 (96%)	2297 (97%)	0.9	690 (97%)	342 (96%)	348 (98%)	0.2
**Systemic therapy performed**	1218 (44%)	144 (40%)	1074 (45%)	0.07	282 (40%)	143 (40%)	139 (39%)	0.8

^1^ Median (IQR); *n* (%); ^2^ Wilcoxon rank sum test; Pearson’s Chi-square test.

**Table 6 cancers-16-00429-t006:** Competing risk regression analyses addressing other-cause mortality (OCM) of newly diagnosed urothelial bladder cancer patients (T1-T4aN0M0) treated with radical cystectomy plus urinary diversion within Surveillance, Epidemiology and End Results (SEER) 2004–2020, after adjustment for cancer-specific mortality (CSM).

	OCM					
	Univariable			Multivariable		
	HR ^1^	95% CI	*p*-Value	HR ^1^	95% CI	*p*-Value
**Ileal conduit**			*Ref*			*Ref*
Abdominal pouch	0.78	0.52–1.15	0.2	0.77	0.52–1.14	0.2
**Age**	1.06	1.02–1.10	**<0.001**	1.05	1.02–1.09	**<0.001**
**Male**			*Ref*			*Ref*
Female	1.31	0.68–2.52	0.4	1.45	0.72–2.91	0.3
**Caucasian**			*Ref*			*Ref*
Non-Caucasian	0.75	0.41–1.35	0.3	0.75	0.41–1.36	0.3
**Married**			*Ref*			*Ref*
Unmarried	0.96	0.6–1.52	0.9	1.02	0.64–1.64	0.9
**T1**			*Ref*			*Ref*
T2	0.76	0.46–1.23	0.3	0.83	0.51–1.37	0.5
T3	0.66	0.36–1.22	0.2	0.72	0.39–1.34	0.3
T4a	1.11	0.52–2.36	0.8	1.25	0.57–2.75	0.6
**Systemic therapy performed**			*Ref*			*Ref*
Not performed	1.19	0.78–1.83	0.4	1.24	0.79–1.95	0.4

^1^ HR = Hazard Ratio, CI = Confidence Interval.

## Data Availability

The data presented in this study are available in this article.

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
