# Peer review of "The Association between Urinary Diversion Type and Other-Cause Mortality in Radical Cystectomy Patients"

_cancers, 2024, doi:10.3390/cancers16020429_

Round 1

Reviewer 1 Report

Comments and Suggestions for Authors

This study conducted with the SEER database wants to identify if complex urinary diversion like abdominal pouch or neobladder are more at risk of other-cancer mortality, than ileal conduit.

The question is interesting but the impact remains limited as there is several bias. The first one is the retrospective design, with limited datas well known with the SEER database, but with the strength of a huge population. Secondly, complex urinary diversion like neobladder or abdominal pouch are usually offered to fit patients, with less comorbidities. Finally, complex reconstruction surgeries usually occur in tertiary centers, with trained surgeons and teams, where it is known that there is a lower mortality rate, regarding to their volume activity.

Several questions:

-          Why the marital status was included in the propensity score matching? How this characteristic would affect the results and mortality? I would have chosen the ECOG status or the Charlson index, more meaningful in this population

-          What do you mean by “systemic therapy”? Is it neo or adjuvant chemotherapy? Immunotherapy?

-          We are also surprised by the fact that there is more abdominal pouch than neobladder in the UD group. How do you explain that?

Author Response

Reviewer #1:

This study conducted with the SEER database wants to identify if complex urinary diversion like abdominal pouch or neobladder are more at risk of other-cancer mortality, than ileal conduit.

The question is interesting but the impact remains limited as there is several bias. The first one is the retrospective design, with limited datas well known with the SEER database, but with the strength of a huge population. Secondly, complex urinary diversion like neobladder or abdominal pouch are usually offered to fit patients, with less comorbidities. Finally, complex reconstruction surgeries usually occur in tertiary centers, with trained surgeons and teams, where it is known that there is a lower mortality rate, regarding to their volume activity.

Response to Reviewer: We would like to express our gratitude for your insightful comments. We agree with the Reviewer that there are several biases using the SEER database. However, we are firmly convinced that the use of retrospective data only marginally limits our analyses, still allowing us to boast an enviable median follow-up of 20 years, which is challenging to achieve in prospective multicenter studies. Regarding the complexity of the reconstruction, we agree with the reviewer that it is mainly offered to patients with fewer comorbidities. However, the SEER does not allow us to consider the patients' comorbidities. In our analyses, we used a population aged between 18 and 70 years to reduce potential complications associated with elderly patients.

Reviewer question: Why the marital status was included in the propensity score matching? How this characteristic would affect the results and mortality? I would have chosen the ECOG status or the Charlson index, more meaningful in this population

Response to Reviewer: We would like to express our gratitude for your insightful comment. We agree with the Reviewer that an ECOG status or a CCI would have improved our propensity score matching. However, these variables are not available in the SEER database. Nevertheless, we included marital status in the matching process, as this factor has proven to be significant in modifying the therapeutic approach in various urological and non-urological malignancies (Marchioni et al. PMID: 28849260; Rosiello et al. PMID: 31468289; Osborne et al. PMID: 16184457). Therefore, the choice of a urinary diversion with the need for a stoma that affects body image could be influenced by the patient's marital status. Furthermore, it should not be overlooked that our outcome of interest is represented by OCM, which is also influenced by deaths due to suicide, with a higher incidence in the never-married population. We improved the limitation section discussing about the ECOG status and the CCI.

Reviewer question: What do you mean by “systemic therapy”? Is it neo or adjuvant chemotherapy? Immunotherapy?

Response to Reviewer: We would like to express our gratitude for your insightful comment. The SEER database does provide information on systemic therapy; however, it does not have the granularity to allow the identification of specific systemic therapy regimens and does not provide information on the number of cycles or duration of its administration. We added this limitation in the limitation section.

Reviewer question: We are also surprised by the fact that there is more abdominal pouch than neobladder in the UD group. How do you explain that?

Response to Reviewer: We would like to express our gratitude for your insightful comments. We were surprised as well for the lower rates of orthotopic neobladder relative to abdominal pouch. However, the results of the current study are consistent with previously published papers on large-scale North American population-based series. Indeed, Nahar et al. (National Cancer Database 2004-2013) recorded among 1736 continent urinary diversion patients, 60% (n=1044) abdominal pouch and 40% (n=698) neobladder patients, respectively. The current study recorded among 628 continent urinary diversion patients, 57% (n=360) abdominal pouch and 43% (n=268) orthotopic neobladder patients, respectively.

Limitation section updated:

Despite its strengths the current study is not devoid of limitations. First, despite a large patient population, the amount of detail is limited. Indeed, only OCM rates were available. Although OCM represents the ultimate marker of life-threatening comorbid-ities and complications, other endpoints would have been of interest. For example, short-term, mid-term, and long-term medical and/or surgical complications according to UD type would ideally be available for purpose of further adjustment. Second, baseline comorbidity status, such as smoking status, body mass index (BMI), diabetes mellitus status, hypertension, cardiovascular disease, the American Society of Anaesthesiologists (ASA) score, as well as ECOG status or Charlson Comorbidity Index (CCI) of RC patients was also not available. Ideally, it could have been used for purpose of further adjustment. Lack of inclusion of medical and/or surgical complications, as well as lack of consideration of baseline comorbidity status, preclude generalizability of our OCM observation re-garding earlier adverse outcomes, such as renal insufficiency, infections, gastrointestinal complications and others that can affect RC patients. Third, adjustments could not be made for patients and surgeon preferences that underlie decision-making regarding UD type. This limitation is applicable to all studies that compare continent UD vs IC [7,18,19]. A randomized design that is free of various biases, such as those described above as well as of confounding of patients and/or surgeon preferences regarding UD type at RC cannot be expected to ever be completed [7,18,19]. Fourth, the amount of details for systemic therapy is limited. Indeed, the SEER database does not have the granularity to allow the identi-fication of specific systemic therapy regimens (chemotherapy or immunotherapy) and does not provide information on cycle number and systemic therapy duration. Fifth, the SEER does not provide any information regarding the surgical approach (open vs lap-aroscopic vs robotic). Indeed, several investigators reported differences in terms of quality of life (QoL) according to surgical approach [20], as well as in the rate of early and late complications that may affect OCM rates [21–23]. Sixth, we only included patients younger than 70 years old. The selection was made in order to minimize differences in comor-bidities in the different groups that may be associated with different OCM rates. Addi-tionally, according to EAR guidelines, orthotopic neobladder should be discouraged in patients older than 80 years old [2]. Moreover, several previous investigators that focused on differences in clinical outcomes according to UD that included orthotopic neobladder recorded a median age between 60 and 70 years old [4,7,24–30]. Therefore, the selection made in the current study reflect the treatment options applied in daily life practice. Finally, the current study shares the limitations of all similar studies that were based on the SEER database and relied on a retrospective data designs[12,18].

Reviewer 2 Report

Comments and Suggestions for Authors

The aim of the study was to compare OCM between ileal conduit and continent UD in patients receiving RC. Topic is interesting, and ideal UD is still debating. However, there are some critical aspects that Authors should review in order to improve the overall quality of manuscript.

- Baseline clinical features should be improved reporting N status, AJCC bladder cancer stage, BMI, smoking status, ASA score. 

- Moreover, Charlson Comorbidity Index or similar must be used in order to compare comorbidities between cohorts. 

- Variables in mregression analysis must be reported such as "female" and not sex relative to male. 

- Surgical approach should be reported. Moreover, discussion could be improved reporting differences between surgical approaches in performing RC and UD. (PMID: 33712389, PMID: 32747981, PMID: 37470132). 

Author Response

Reviewer #2:

The aim of the study was to compare OCM between ileal conduit and continent UD in patients receiving RC. Topic is interesting, and ideal UD is still debating. However, there are some critical aspects that Authors should review in order to improve the overall quality of manuscript.

Reviewer question: Baseline clinical features should be improved reporting N status, AJCC bladder cancer stage, BMI, smoking status, ASA score. 

Response to Reviewer: We would like to express our gratitude for your insightful comments. The SEER database does not report BMI, smoking status and ASA score. We added this limitation in the limitation section. All the patients included in the current manuscript harbored T2-T4aN0M0. Therefore, we decided to report the T stage instead of AJCC because according to the AJCC the group IIIa would have included also N1 patients.

Reviewer question: Moreover, Charlson Comorbidity Index or similar must be used in order to compare comorbidities between cohorts. 

Response to Reviewer: We would like to express our gratitude for your insightful comment. However, the SEER database does not report Charlson Comorbidity Index. We added this limitation in the limitation section.

Reviewer question: Variables in regression analysis must be reported such as "female" and not sex relative to male. 

Response to Reviewer: We would like to express our gratitude for your insightful comment. We corrected accordingly.

Reviewer question: Surgical approach should be reported. Moreover, discussion could be improved reporting differences between surgical approaches in performing RC and UD. (PMID: 33712389, PMID: 32747981, PMID: 37470132). 

Response to Reviewer: We would like to express our gratitude for your insightful comment. The SEER database does not report the surgical approach. Therefore we added this limit in the limitation section, highlighting the importance of surgical approach in terms of quality of life, as well as clinical outcomes.

Limitation section updated:

Despite its strengths the current study is not devoid of limitations. First, despite a large patient population, the amount of detail is limited. Indeed, only OCM rates were available. Although OCM represents the ultimate marker of life-threatening comorbid-ities and complications, other endpoints would have been of interest. For example, short-term, mid-term, and long-term medical and/or surgical complications according to UD type would ideally be available for purpose of further adjustment. Second, baseline comorbidity status, such as smoking status, body mass index (BMI), diabetes mellitus status, hypertension, cardiovascular disease, the American Society of Anaesthesiologists (ASA) score, as well as ECOG status or Charlson Comorbidity Index (CCI) of RC patients was also not available. Ideally, it could have been used for purpose of further adjustment. Lack of inclusion of medical and/or surgical complications, as well as lack of consideration of baseline comorbidity status, preclude generalizability of our OCM observation re-garding earlier adverse outcomes, such as renal insufficiency, infections, gastrointestinal complications and others that can affect RC patients. Third, adjustments could not be made for patients and surgeon preferences that underlie decision-making regarding UD type. This limitation is applicable to all studies that compare continent UD vs IC [7,18,19]. A randomized design that is free of various biases, such as those described above as well as of confounding of patients and/or surgeon preferences regarding UD type at RC cannot be expected to ever be completed [7,18,19]. Fourth, the amount of details for systemic therapy is limited. Indeed, the SEER database does not have the granularity to allow the identi-fication of specific systemic therapy regimens (chemotherapy or immunotherapy) and does not provide information on cycle number and systemic therapy duration. Fifth, the SEER does not provide any information regarding the surgical approach (open vs lap-aroscopic vs robotic). Indeed, several investigators reported differences in terms of quality of life (QoL) according to surgical approach [20], as well as in the rate of early and late complications that may affect OCM rates [21–23]. Sixth, we only included patients younger than 70 years old. The selection was made in order to minimize differences in comor-bidities in the different groups that may be associated with different OCM rates. Addi-tionally, according to EAR guidelines, orthotopic neobladder should be discouraged in patients older than 80 years old [2]. Moreover, several previous investigators that focused on differences in clinical outcomes according to UD that included orthotopic neobladder recorded a median age between 60 and 70 years old [4,7,24–30]. Therefore, the selection made in the current study reflect the treatment options applied in daily life practice. Finally, the current study shares the limitations of all similar studies that were based on the SEER database and relied on a retrospective data designs[12,18].

Reviewer 3 Report

Comments and Suggestions for Authors

This study was aimed to compare OCM between ileal conduit and continent UD in patients receiving RC. UD selection has always been challenging and studies comparing differences between UDs and comparing survival are always of great interest, therefore Authors should be commended. 

- Baseline clinical features of both cohorts must be improved. Particularly, comorbidities must be fully reported being strictly related to OCM. Therefore, comorbidities index should be reported. Otherwise, diabetes, BMI, hypertension, cardiovascular disease should be included. 

- Surgical approach should be included and Authors should specify surgical approach for both RC and UD. Discussion should be improved with recent evidences supporting differences between surgical approach (PMID: 34986007). 

- Authors included only patients younger than 70yrs old. This aspect should be further discussed. 

Author Response

Reviewer #3:

This study was aimed to compare OCM between ileal conduit and continent UD in patients receiving RC. UD selection has always been challenging and studies comparing differences between UDs and comparing survival are always of great interest, therefore Authors should be commended. 

Reviewer question: Baseline clinical features of both cohorts must be improved. Particularly, comorbidities must be fully reported being strictly related to OCM. Therefore, comorbidities index should be reported. Otherwise, diabetes, BMI, hypertension, cardiovascular disease should be included. 

Response to Reviewer: We would like to express our gratitude for your insightful comment. However, the SEER database does not report Charlson Comorbidity Index, as well as diabetes status, BMI, hypertension or cardiovascular disease. We added this limitation in the limitation section.

Reviewer question: Surgical approach should be included and Authors should specify surgical approach for both RC and UD. Discussion should be improved with recent evidences supporting differences between surgical approach (PMID: 34986007). 

Response to Reviewer: We would like to express our gratitude for your insightful comment. The SEER database does not report the surgical approach. Therefore we added this limit in the limitation section, highlighting the importance of surgical approach in terms of quality of life, as well as clinical outcomes.

Reviewer question: Authors included only patients younger than 70yrs old. This aspect should be further discussed. 

Response to Reviewer: We would like to express our gratitude for your insightful comments. The decision to include only patients under the age of 70 in our analyses is grounded in several considerations. First and foremost, our primary objective was to identify differences in terms of OCM, which can be influenced by the age of patients. By excluding the elderly population, any observed disparities would primarily stem from variations in the urinary diversion approach employed. Furthermore, in accordance with the EAU guidelines, the creation of an orthotopic bladder is discouraged in patients above 80 years of age (although not explicitly mandated). This aligns with established medical practices. Lastly, a review of the existing literature on orthotopic neobladders reveals that the median age typically falls between 60 and 70 years in the majority of studies. We are aware that this decision may introduce limitations to our study, and we have duly acknowledged this in the limitation section of our work.

Limitation section updated:

Despite its strengths the current study is not devoid of limitations. First, despite a large patient population, the amount of detail is limited. Indeed, only OCM rates were available. Although OCM represents the ultimate marker of life-threatening comorbid-ities and complications, other endpoints would have been of interest. For example, short-term, mid-term, and long-term medical and/or surgical complications according to UD type would ideally be available for purpose of further adjustment. Second, baseline comorbidity status, such as smoking status, body mass index (BMI), diabetes mellitus status, hypertension, cardiovascular disease, the American Society of Anaesthesiologists (ASA) score, as well as ECOG status or Charlson Comorbidity Index (CCI) of RC patients was also not available. Ideally, it could have been used for purpose of further adjustment. Lack of inclusion of medical and/or surgical complications, as well as lack of consideration of baseline comorbidity status, preclude generalizability of our OCM observation re-garding earlier adverse outcomes, such as renal insufficiency, infections, gastrointestinal complications and others that can affect RC patients. Third, adjustments could not be made for patients and surgeon preferences that underlie decision-making regarding UD type. This limitation is applicable to all studies that compare continent UD vs IC [7,18,19]. A randomized design that is free of various biases, such as those described above as well as of confounding of patients and/or surgeon preferences regarding UD type at RC cannot be expected to ever be completed [7,18,19]. Fourth, the amount of details for systemic therapy is limited. Indeed, the SEER database does not have the granularity to allow the identi-fication of specific systemic therapy regimens (chemotherapy or immunotherapy) and does not provide information on cycle number and systemic therapy duration. Fifth, the SEER does not provide any information regarding the surgical approach (open vs lap-aroscopic vs robotic). Indeed, several investigators reported differences in terms of quality of life (QoL) according to surgical approach [20], as well as in the rate of early and late complications that may affect OCM rates [21–23]. Sixth, we only included patients younger than 70 years old. The selection was made in order to minimize differences in comor-bidities in the different groups that may be associated with different OCM rates. Addi-tionally, according to EAR guidelines, orthotopic neobladder should be discouraged in patients older than 80 years old [2]. Moreover, several previous investigators that focused on differences in clinical outcomes according to UD that included orthotopic neobladder recorded a median age between 60 and 70 years old [4,7,24–30]. Therefore, the selection made in the current study reflect the treatment options applied in daily life practice. Finally, the current study shares the limitations of all similar studies that were based on the SEER database and relied on a retrospective data designs[12,18].

Round 2

Reviewer 2 Report

Comments and Suggestions for Authors

No further comments 

Reviewer 3 Report

Comments and Suggestions for Authors

No further comments